# Numerical Simulation Study on the Layered Phenomenon of Lakes and Reservoirs Destroyed by a Forced Circulation Device

**Jiaxing Xu** [1], **Minghan Luo** [1,2,*], **Dema Ba** [3], **Gongde Wu** [1,2], **Ligang Xu** [4] and **Taeseop Jeong** [5]

1 School of Environmental Engineering, Nanjing Institute of Technology, Nanjing 211167, China; xujiaxing1987@163.com (J.X.); wugongde@njit.edu.cn (G.W.)
2 Energy Research Institute, Nanjing Institute of Technology, Nanjing 211167, China
3 Alxa Ecological Environment Monitoring Station, Alxa League 750306, China; badema0483@163.com
4 State Key Laboratory of Lake Science and Environment, Nanjing Institute of Geography and Limnology, Chinese Academy of Sciences, Nanjing 210008, China; lgxu@niglas.ac.cn
5 Department of Environmental Engineering, Jeonbuk National University, Jeonju 561-756, Korea; jeongts@jbnu.ac.kr
* Correspondence: leon96201@njit.edu.cn

**Abstract:** Thermal stratification is a common phenomenon in lakes and reservoirs. It has a significant influence on water quality dynamics. The destruction of the thermal stratification of lakes and reservoirs can affect the water environment, improve the water quality and the water environment quality and prevent the occurrence of eutrophication. In this study, computational fluid dynamics (CFD) combined with a Eulerian two-phase flow model is used to predict the damage caused by an airlift device to the thermal stratification phenomenon of lake water. The results show that the two devices with different sizes can contribute to a certain exchange of kinetic and thermal energy, affecting the liquid velocity and temperature in the lake water under the condition of different gas velocities. Furthermore, the degree of damage to the thermal stratification phenomenon of lake reservoir is small. However, with the same gas velocity, the device with a guide plate can have a greater impact on the liquid velocity and temperature in the lake water. Further prediction results show that the airlift installed with a guide plate can affect the heat transfer of liquid in the lake and reservoir better and destroy the thermal stratification phenomenon effectively. The quantitative results of model prediction can provide an effective basis for future field scale-up experiments.

**Keywords:** stratified reservoir; CFD; airlift; lake; reservoir

## 1. Introduction

At present, the surface water resources of all countries in the world are facing the same water environment problem, which is water pollution. Once the water resources are polluted, the reserves of available water resources decrease sharply, thereby causing great inconvenience to the development of cities and the lives of citizens, and even directly threatening the ecological environment and people's lives. The thermal stratification of reservoirs and lakes, which account for most of the surface water, are amongst the main reasons for the deterioration of water quality due to seasonal differences [1,2]. Most lakes and reservoirs usually experience the seasonal thermal stratification of water bodies with great water depth, a long hydraulic retention time and poor flow mobility, which changes the natural runoff conditions of the original water flow, leads to the destruction of the ecological balance of the water body and damages the water quality's natural repair and adjustment function, thereby causing a series of water quality problems. The occurrence of thermal stratification in the water body hinders the vertical transmission function of dissolved oxygen, and the double oxygen consumption of the water body and the sediment causes the lower water body to be prone to an anaerobic state during thermal stratification, thereby resulting in the death of aquatic organisms due to a lack of



oxygen. This occurrence also promotes the release of nutrients and metal substances, such as TN, TP, iron, manganese, and sulphide, in the sediments to the upper water body [3]. The enriched macronutrients in the upper water body are conducive to the growth of toxic algae and the proliferation of plankton, which ultimately causes 'eutrophication'. The periodic mass propagation of algae caused by eutrophication will greatly reduce the transparency of water bodies, consume dissolved oxygen in the water and disrupt the balance of aquatic ecosystems. Eventually, the pollution of lakes and reservoirs will become increasingly serious, thereby harming the aquatic ecosystems and endangering the safety of drinking water.

The thermal stratification structure of the water body is related to many factors. The seasonal variation of the thermal stratification structure of different lakes and reservoirs is obvious. Generally, the greater the depth of the lake is, the more stable the stratification will be [4–6]. Hasan et al. explored lakes and reservoirs in tropical regions and found that the thermal stratification stability of water bodies in the late summer and early autumn seasons was still relatively large, and the mixing process was relatively slow [7]. Bonnet et al. observed the Villerest Reservoir in France for two years and concluded that it was in thermal stratification from April to September [8]. Water temperature is not only the main water quality indicator of the water body, but is also the main factor that affects other water quality indicators. The thermal stratification structure caused by water temperature has a direct impact on the physical, chemical and biological processes of substances in the water. The resulting density difference hinders the vertical mixing of the water body and affects the vertical migration and diffusion of dissolved oxygen and nutrients in the water. Notably, destroying the radial thermal stratification of the lake reservoir is an effective way to improve the deterioration of water quality.

In recent decades, many researchers have carried out considerable research on destroying the thermal stratification of lakes and reservoirs. Arlo et al. and Gerling et al. applied oxygenation devices to study the oxygenation effect of lake water, and the results showed that the dissolved oxygen in the constant temperature layer could reach the supersaturation state but the thermal layered structure of the water body was not destroyed [9,10]. The constant-temperature layer aeration technology has a high oxygenation efficiency for the lower water body and can maintain the lower layer's aquatic ecology. However, it has no control effect on algae outbreaks because it cannot achieve full-layer mixing. Studies found that the constant-temperature layer oxygenation technology will increase the oxygen demand of the water body and sediments in the constant-temperature layer [11,12], which requires the equipment to operate for a long time, resulting in higher operating costs. Sanjina et al. studied algae control and mixing effects in Falling Creek Reservoir using a solar-driven mechanical mixing method [13]. The results showed that the operation of the device controlled the algal bloom and weakened the thermal stratification structure of the water body effectively. Such aeration and mechanical mixing methods are currently the most widely used and most effective physical repair technologies. However, their engineering volume and investment are relatively large, they can only repair water bodies in a short period of time, and maintaining the quality of the water environment effectively in the long term is difficult. In addition, with the rapid development of computer methods and performance, many researchers have used commercial software to simulate the flow patterns, water temperature and water quality ecology of lakes and reservoirs of different scales. Lee et al. simulated the water temperature structure of Daecheong Reservoir in South Korea, which has relatively deep water, and predicted the thermal stratification structure of the reservoir using the ELCOM model successfully [14]. Seo et al. used the EFDC model to simulate the water temperature structure of Yongdam Lake and predicted the dynamic changes in water level fluctuations and thermal stratification successfully [15]. Li et al. and Li et al. based on multi-year in situ water quality measurements and meteorological data during natural and artificial mixing periods, evaluated the effects of water-lifting aerators and climatic factors on convective mixing processes and their duration [16,17].

The methods that can be used to solve the thermal stratification phenomenon of lakes and reservoirs can be diversified. However, no reports about the application of the airlift method to the thermal stratification phenomenon of lakes and reservoirs are available. This article attempts to use the airlift method to forcefully destroy the thermal stratification phenomenon of lake and reservoirs so that the water body can continue to maintain a vertical circulation state. To obtain the experimental conditions in the field, the computational fluid dynamics (CFD) method was used to predict the influence of airlift devices with different sizes and gas speeds on the fluid velocity and temperature changes in the lake water. Finally, the optimal conditions for the airlift device to operate in the water body were analyzed.

## 2. Numerical Methods

### 2.1. Simulated Domain and Geometry

A flow was used by continuously operating the airlift device to experimentally evaluate the CFD results. The two-dimensional geometry and configuration for the airlift device in the lake are shown in Figure 1. The geometry was created in ANSYS Design Modeler software. Given the characteristics of the symmetric structure of the device, the flow field simulation could be reasonably simplified by a 2D axial symmetry model. An unstructured numerical grid was implemented with 8662 elements, where the min size of the grid was 0.15 mm, and the max face size was 5.0 mm.

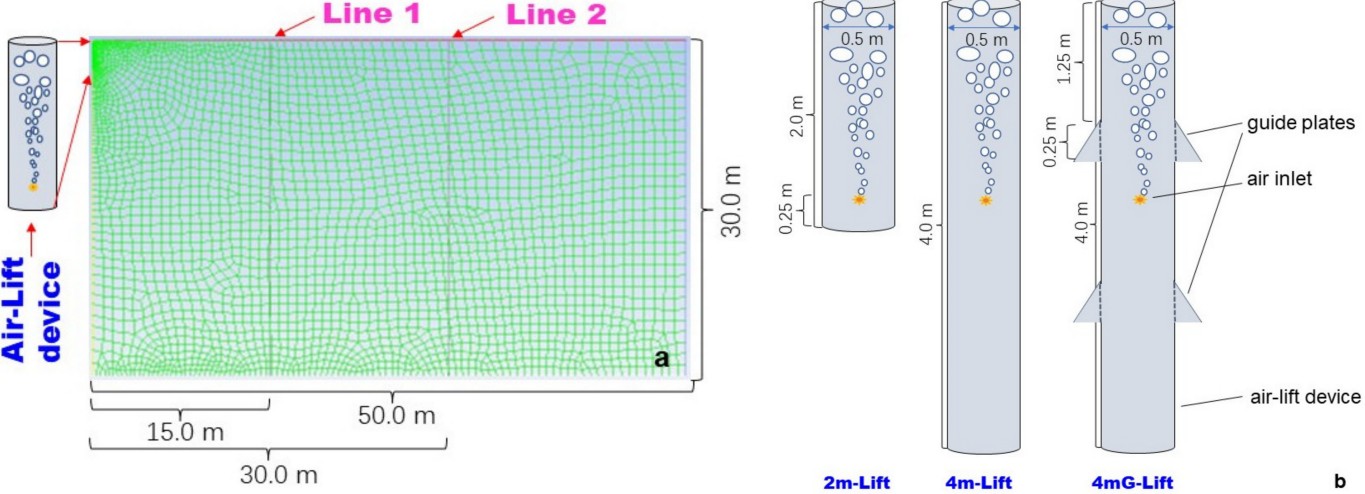

**Figure 1.** Schematic of the location of airlift device in lake water (**a**) and different configuration of the device (**b**).

The range of the lake water designed in this study, with the depth of 30 m and width of 50 m, is shown in Figure 1a. To better observe the destruction of airlift device to the thermal stratification phenomenon in the lake water, two longitudinal data lines, namely Lines 1 and 2, were set at 15 m and 30 m away from airlift in the lake water, respectively. The airlift device was composed of a simple pipe with a diameter of 0.5 m, as shown in Figure 1b, and its air inlet was installed in the central position 1.75 m down from the upper part of the cylinder. The lengths of the airlifts were 2 m and 4 m, and the marks of the three lift types were 2m-Lift, 4m-Lift and 4mG-Lift. For the 4 mG lift, two guide plates are installed on the 4 m long airlift device.

### 2.2. Initial and Boundary Conditions

For the initial conditions, the gas phase entered the lake from the bottom of the airlift at different flow rates. The gas phase in the standard state was air. A uniform air velocity $U_g$ (0.03, 0.07, 0.10 and 0.15 m/s) was simulated from the gas inlet. The considered diameter of

air was 0.2 mm. For the airlift wall boundary, the gas phase was treated as free-slip [13,18]. A no-slip boundary condition was imposed on the walls. In addition, Figure 2 shows that the initial condition of the thermal stratification temperature in the lake water was set as 277–303 K with the increase in water depth.

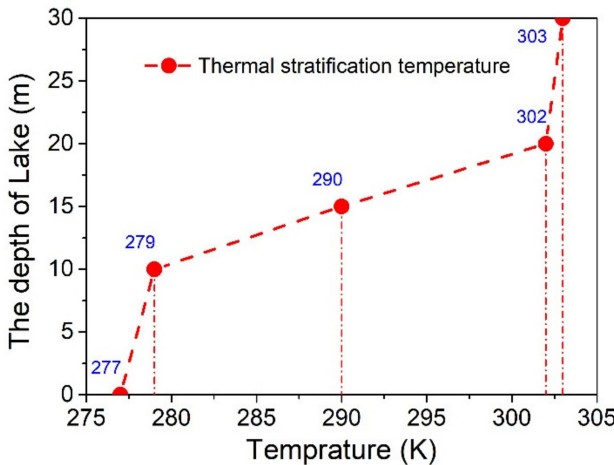

**Figure 2.** Temperature settings of thermal stratification of lake water (K).

### 2.3. Numerical Solution

ANSYS Fluent 16.2 was used to read the mesh and perform the CFD computations. The segregated unsteady state solver was used to solve the governing equations. The SIMPLE algorithm was selected for the pressure–velocity coupling. Second-order upwind discretisation scheme was applied, except for pressure, for which the standard scheme was selected. The variation of velocity magnitude at several points of the computational domain was used as an indicator of convergence (at least 20 iterations) [19,20]. Although simulation is always tracked with time, the solution algorithm was run with steady and transient flow simulations.

## 3. Results and Discussion

### 3.1. Velocity Distribution

An airlift lake water circulation device refers to a device that causes a density difference between the liquid inside and outside the diversion pipe by aeration, so that the liquid can circulate from the inside to the outside continuously. This continuous circulation can bring power to the static (vertical and horizontal) lake water and force the destruction of stratification to prevent eutrophication. To investigate the influence of an airlift lake water circulation device on stratification phenomenon under different gas velocity conditions, the velocity field in the whole lake water was analyzed initially, and then the data at 15 m (line 1) and 30 m (line 2) away from the circulation device were selected for analysis and research.

Figure 3 shows the fluid velocity distribution of the different circulation devices (e.g., 2m-Lift, 4m-Lift and 4mG-Lift) under different gas velocities (e.g., 0.03, 0.07, 0.10 and 0.15 m/s). The airlift lake water circulation device mainly differentiates the liquid density inside and outside the diversion pipe by aeration to naturally produce the vertical circulation of lake water. The range of fluid velocity shown in the figure is 0–0.01 m/s. For the 2m-Lift and 4m-Lift circulating devices, the liquid velocity along the water depth increases gradually and obviously, whilst the liquid velocity along the water surface decreases gradually. Regarding the range of velocity, a liquid velocity of more than 0.01 m/s is concentrated around the airlift circulation device, and its influence range is approximately 1000 m³ (8 m × 8 m × 15 m). Meanwhile, for the 4m-Lift, the influence range gradually increases with the gas velocity. The stratification of liquid in the lake water can be greatly affected by 7–8 m around the airlift circulation device and 15 m downwards. In addition, for the distribution of liquid velocity less than 0.01 m/s, the area with a higher velocity is

distributed from the middle of the lake to the bottom downwards, and it can be affected to approximately 30 m in the transverse direction, and it can be affected to about 40 m for 4m-Lift. This phenomenon is due to the fact that the airlift lake water circulation device improves the liquid with low density in the guide tube by aeration. At this time, the liquid at the bottom of the guide tube quickly moves upward from the bottom of the lake water and increases the speed of movement to supplement the liquid lifted in the guide tube in time. Therefore, the airlift lake water circulation device in this study can lift the constant-temperature water layer at the bottom of the lake to the upper variable-temperature layer area effectively and finally achieve the purpose of forcibly destroying the temperature stratification of the lake water.

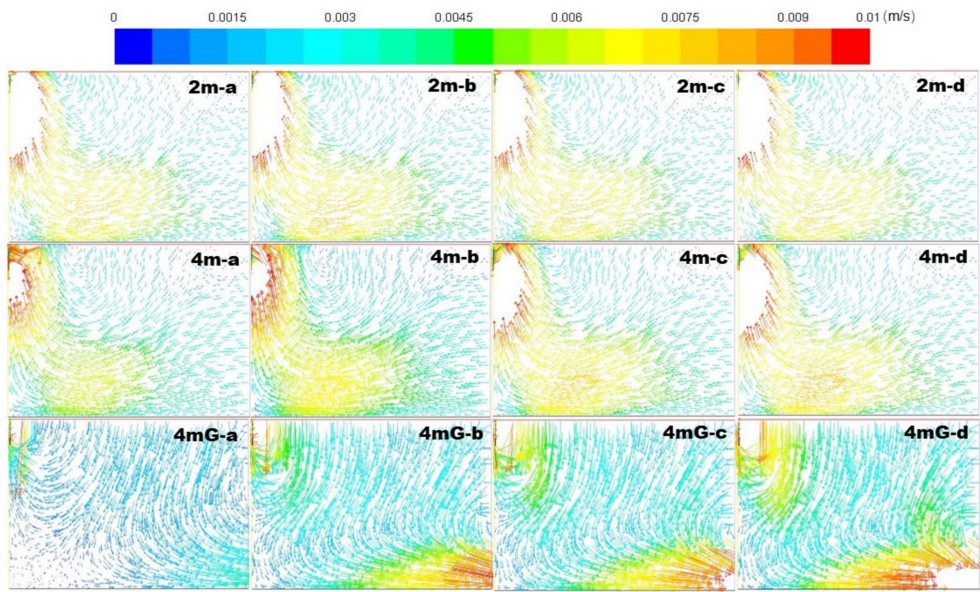

**Figure 3.** Velocity distribution of different gas inlet velocities (e.g., a, b, c and d are expressed as 0.03, 0.07, 0.10 and 0.15 m/s, respectively) produced by different lifts (e.g., 2m, 4m and 4mG are expressed as 2m-Lift, 4m-Lift, and 4mG-Lift).

According to the liquid velocity distribution of the 2m-Lift and 4m-Lift circulation devices, with the extension of the guide tube, the influence range of the liquid velocity in the lake water can be increased effectively. However, from the numerical simulation results in this study, only the liquid velocity around the circulation device is larger, and the liquid velocity along the water surface gradually decreases. Thus, the 4mG-Lift circulation device is further improved. The velocity distribution in the lake water of the 4mG-Lift unit is different from those of the 2m-Lift and 4m-Lift units without guide plates. When guide plates are installed, the liquid in the lower and middle parts of the guide tube can enter the guide tube, which makes the liquid flow velocity in the whole lake more uniform. The 4mG-Lift circulation device has a greater influence on the liquid velocity in the lake water, which is more extensive and uniform. It can maintain a larger liquid velocity distribution along the water surface or towards the bottom. Moreover, with the increase in gas velocity, the velocity of the liquid flow in the constant temperature layer and the disturbance, which is conducive to the mixing effect between the upper and lower water flows and destroys the stratification phenomenon, increases effectively.

Table 1 shows the maximum velocity distribution of different types of circulation devices under different gas velocity conditions. The velocity of the 2m-Lift type circulating device is the largest, and it increases with the gas velocity. However, Figure 3 shows that the liquid flow velocity of the 2m-Lift type circulating device greater than 0.01 m/s appears in the range of approximately 8 m around the circulating device, and the liquid flow velocity far away from the device is low. Although the maximum velocity of the 4mG-Lift circulation device is low, the influence range of liquid velocity can be extended to

about 50 m horizontally and 30 m vertically with the increase in gas velocity. In addition, the biggest advantage of the 4mG-Lift is that it can produce a larger liquid flow velocity in the opposite direction of the circulating device in the water depth, thereby bringing power to the liquid flow of the constant temperature layer.

**Table 1.** Maximum velocity with different gas velocities produced by different lifts (m/s).

| Inlet Velocity | 2m-Lift | 4m-Lift | 4mG-Lift |
|---|---|---|---|
| 0.03 m/s | 0.478 | 0.436 | 0.374 |
| 0.07 m/s | 0.698 | 0.650 | 0.509 |
| 0.10 m/s | 0.799 | 0.786 | 0.579 |
| 0.15 m/s | 0.930 | 0.906 | 0.670 |

*3.2. The Temperature Distribution*

Considering that the airlift lake water circulation device can produce the circulation flow phenomenon to the longitudinal direction of the lake water, it brings power to the constant temperature and thermocline layers and then forcibly destroys the phenomenon of water temperature stratification. Figure 4 shows the influence of different circulation devices (e.g., 2m-Lift, 4m-Lift and 4mG-Lift) on the temperature distribution of lake water under various gas velocities (e.g., 0.03, 0.07, 0.10 and 0.15 m/s). In this study, the temperature range of the lake water is artificially set at 277–303 K along the water depth direction (Figure 1), and the boundary conditions of the lake water are set to the mode of continuous energy exchange through user-defined software. According to the damage caused by the 2m-Lift and 4m-Lift circulation devices to water temperature stratification, different gas velocities have a certain damage effect on the phenomenon of water temperature stratification. However, the difference is small, and the influence range can reach approximately 30 m in the radial direction. Meanwhile, a constant temperature layer beyond 30 m cannot be damaged. For the 4mG-Lift cycle device, the water temperature stratification can be destroyed under different gas velocity conditions, which is better than the two other devices. Compared with the two other devices, 4mG-Lift cycle device has an obvious destructive effect on the constant temperature layer at the bottom of the lake, and its radial influence range can reach more than 50 m. However, when the gas velocity is less than 0.03 m/s, the overall water temperature distribution is more uniform, because the density difference between the inside and outside of the guide tube caused by the gas velocity at this time causes the liquid to flow into the bottom of the guide tube through the guide plate to reach a certain degree of balance with the flow at the upper outlet. This instance also shows that the guide plate should be added to the 4m lift device. The velocity and flow rate of liquid circulation in the lake water are increased with the gas velocity. Meanwhile, the fixed diameter of the guide tube cannot produce a greater flow rate effectively. Hence, most of the liquid velocity is retained at the bottom of the lake, so a higher liquid velocity is generated at the bottom of the lake in the opposite direction of the circulation device, which is consistent with the liquid velocity distribution in Figure 3.

*3.3. Vertical Distribution of Water Temperature and Velocity*

The fact that the airlift lake water circulation device can destroy the stratification phenomenon of the lake water is confirmed by the velocity distribution and the change in water temperature in the lake water. However, a quantitative standard is needed for the change in velocity and temperature in the lake water. Figures 4 and 5 show the radial distribution of velocity and temperature with the gas velocity of the 2m-Lift and 4m-Lift circulation devices in lake water, whilst (a) and (b) of Figures 4 and 5 show the data at 15 m (Line 1) and 30 m (Line 2) away from the guide tube, respectively. With the increase in gas velocity, the liquid velocity at Lines 1 and 2 show a gradually increasing trend in the vertical direction, and the maximum velocity can reach 0.08 m/s and 0.06 m/s, and the maximum velocity also appears at the bottom of the lake, thereby helping to destroy the water temperature of the constant temperature layer. As shown in Figure 6, the liquid

velocity at Line 1 of the 4m-Lift cycle shows a great difference with the increase in gas velocity. Under different gas velocity conditions, the change in temperature does not show a great difference. However, the overall temperature of Line 1 shows a quasilinear trend. Compared with the initial temperature (i.e., 277–303 K), the maximum temperature at the bottom of the lake reaches approximately 282 K, indicating that the horizontal and vertical ranges of 15 m and 30 m can affect the thermal stratification of the lake. However, a big difference between the constant temperature layer and the variable temperature layer of Line 2 at a transverse distance of 30 m is still observed, and the mixing effect between the upper and lower water layers is insufficient.

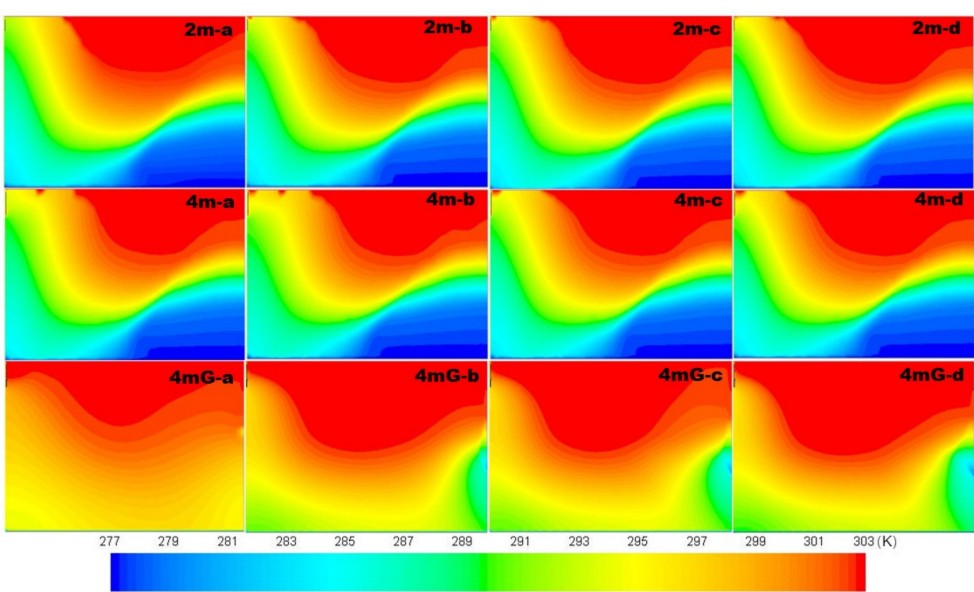

**Figure 4.** Temperature distribution of different gas inlet velocities. (e.g., a, b, c and d are expressed as 0.03, 0.07, 0.10 and 0.15 m/s, respectively) produced by different lifts (e.g., 2m, 4m and 4mG are expressed as 2m-Lift, 4m-Lift, and 4mG-Lift).

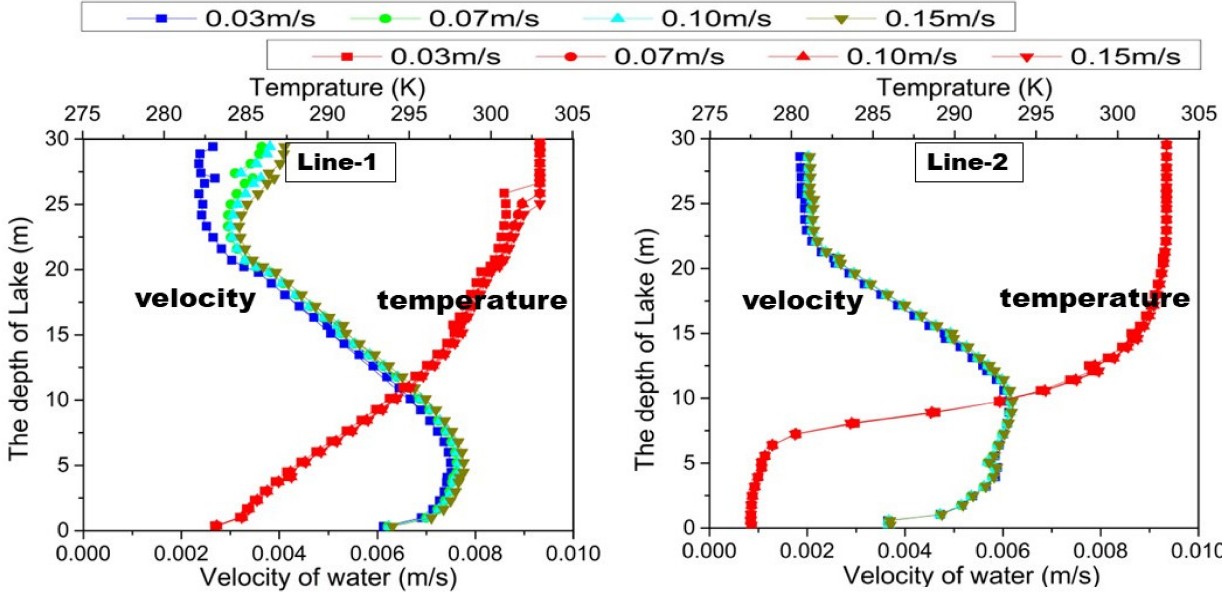

**Figure 5.** Vertical distribution of water temperature and velocity with different gas inlet velocities in Line 1 and Line 2 produced by 2m-Lift.

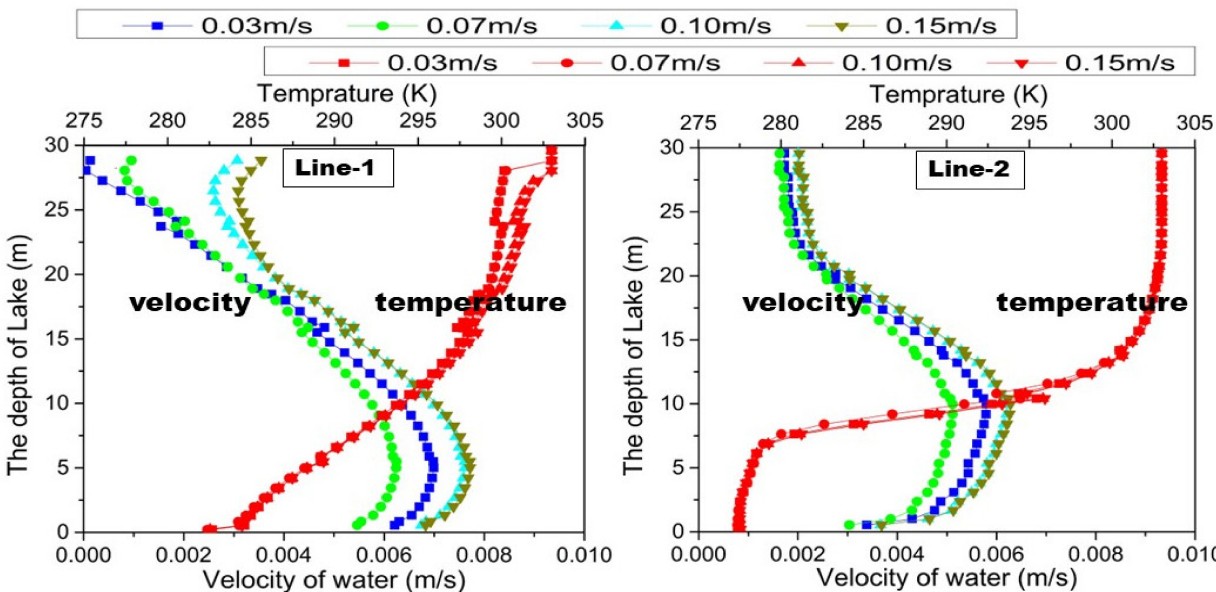

**Figure 6.** Vertical distribution of water temperature and velocity with different gas inlet velocities in Line 1 and Line 2 produced by 4m-Lift.

In this study, a guide plate was added to the 4m-Lift cycle and compared with the previous device. Figure 7 shows the effect of the 4mG-Lift cycle on liquid velocity and water temperature in lake water under different gas velocity conditions. Some differences in the longitudinal velocity changes of the two sections (a and b) under different gas velocity conditions were observed. In particular, the liquid velocity of Line 1 shows an increasing trend with the increase in gas velocity, whereas the liquid velocity of Line 2 shows a small difference because it is far away from the guide tube. The longitudinal water temperature changes of the two sections show that the difference is larger than the initial water temperature distribution (277–303 K). Regardless of whether it is Line 1 or Line 2, the maximum water temperature at the bottom of the lake reached 296 K, indicating that the overall water temperature range narrowed to 296–303 K. The thermal stratification phenomenon between the upper and lower water temperatures almost disappeared. This observation shows that compared with other types of circulation devices, the 4mG-Lift circulation device can effectively increase the vertical or horizontal circulation effect of lake water, promote the heat exchange between water layers and finally achieve the destruction of the thermal stratification phenomenon of lake water. The use of this airlift device can effectively destroy the water thermal stratification phenomenon of the lake and prevent the eutrophication problem further. However, considering the problems of energy consumption and investment cost and the changes in liquid velocity and longitudinal temperature, this study suggests that the gas velocity in the device can be controlled at approximately 0.07 m/s.

### 3.4. Hydrodynamics of 4mG-Lift

The numerical simulation of the liquid dynamic performance of airlift devices in lake water based on a CFD model is shown in Figure 8. The simulation of this study aims to simulate the hydrodynamic distribution after the steady state in an unsteady environment. The inherent physical characteristics of an airlift can enable the liquid outside the guide tube to enter the bottom of the guide tube and flow out from the top because of the density difference. This continuous hydrodynamic process can produce disturbances in the lake water and finally bring kinetic energy to the relatively still water. Although the 2m-Lift and 4m-Lift devices operate in the lake water, they can bring some kinetic energy to the horizontal or vertical liquid continuously and exchange the temperature through the mixing effect between the water layers to destroy the thermal stratification. However, the

circulating water is not large enough in area to transfer kinetic energy further. Therefore, the installation of two guide plates on the original 4m-Lift device can increase the water flow of the device effectively. The hydrodynamic phenomenon is shown in the figure. The circulating water caused by the density difference inside and outside the device can enter from different directions of the device, thereby transferring the kinetic energy to the liquid farther away.

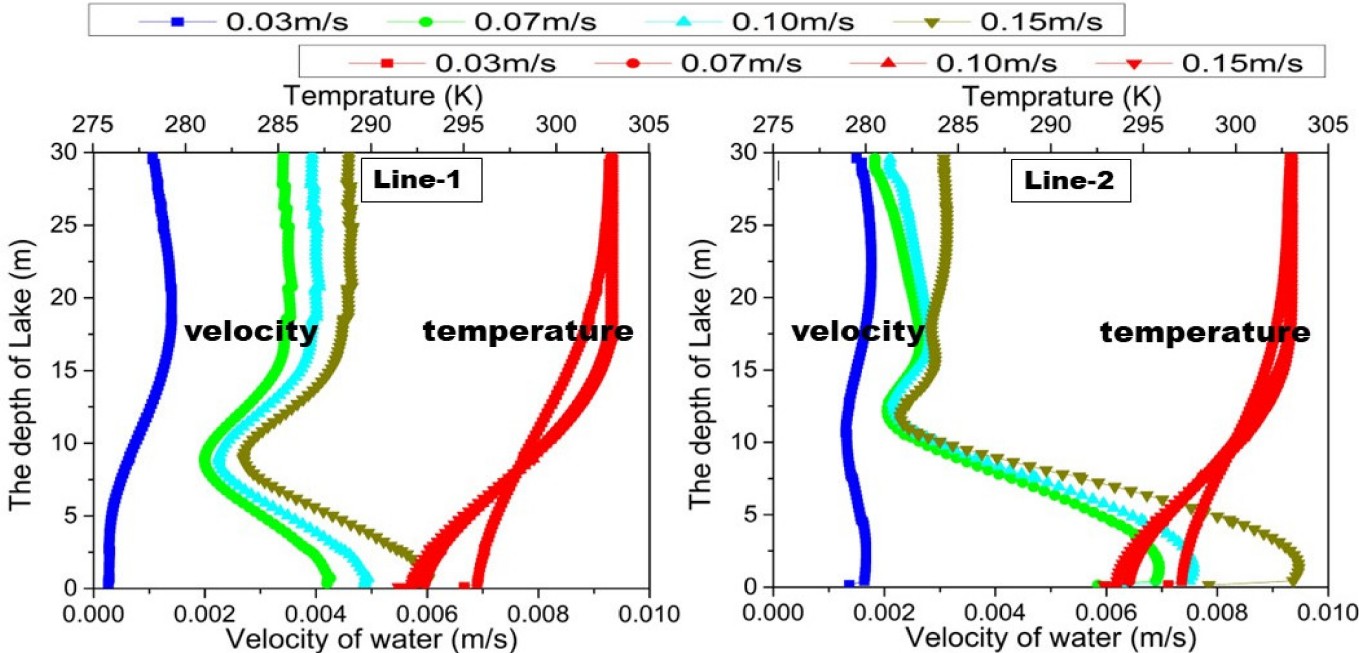

**Figure 7.** Vertical distribution of water temperature and velocity with different gas inlet velocities in Line 1 and Line 2 under 4mG-Lift.

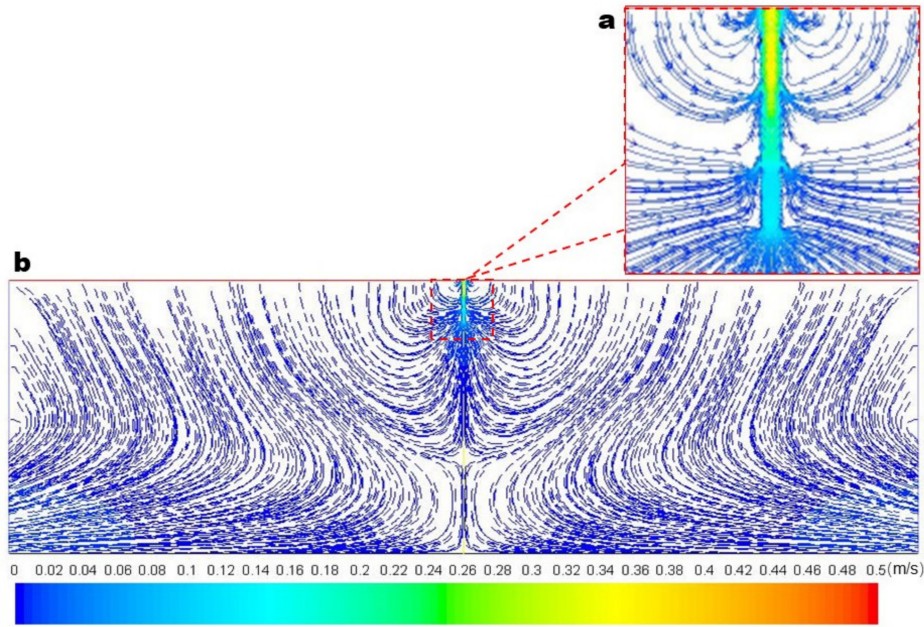

**Figure 8.** Hydrodynamic analysis of the 4mG-Lift device in lake water. (**a**) Hydrodynamics around the 4mG-Lift; (**b**) hydrodynamics of the whole lake.

## 4. Conclusions

In this paper, the Euler model is used to simulate the hydrodynamic phenomena of an airlift device in the lake water, and the mechanical behavior of the liquid in the water is analyzed with different sizes and in different directions, and the damage caused to the thermal stratification of the lake water under different gas velocities is discussed. As seen in the simulation results of the flow field, the liquid driven by the gas velocity can drive the lake water into the circulation state more effectively, transfer the kinetic energy effectively, increase the mixing effect between the lake water layers, destroy the thermal stratification phenomenon and finally improve the eutrophication of the lake water. The model used in this study can not only describe the hydrodynamic behavior of airlift devices in lake water accurately, but also further quantify the relevant physical parameters in the liquid, such as velocity and temperature, which can provide a favorable reference for the future application of airlifts in addressing the thermal stratification of lake water.

**Author Contributions:** Methodology, L.X. and T.J.; software, L.X., M.L. and T.J.; visualization, J.X. and M.L.; formal analysis, J.X., M.L., L.X. and T.J.; investigation, J.X. and G.W.; resources, G.W.; data curation, J.X., M.L. and D.B.; writing—original draft preparation, J.X. and M.L.; writing—review and editing, J.X., M.L., L.X. and T.J.; validation, D.B. and L.X.; supervision, D.B.; project administration, J.X. and M.L.; funding acquisition, M.L. All authors have read and agreed to the published version of the manuscript.

**Funding:** This work was supported by the National Key Research and Development Program of China (grant number 2018YFE0206400); the National Natural Science Foundation of China (grant number 41971137, U2240224); Introduction Talent Scientific Research Foundation Project of NanJing Institute of Technology (grant number YKJ201846); Supported by the Cooperation Fund of Energy Research Institute, Nanjing Institute of Technology (No. CXY201925).

**Institutional Review Board Statement:** Not applicable.

**Informed Consent Statement:** Not applicable.

**Data Availability Statement:** Not applicable.

**Acknowledgments:** We are grateful to the editor and anonymous reviewers for their efforts to improve this manuscript.

**Conflicts of Interest:** The authors declare no conflict of interest.

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
