# Peer review of "Numerical Simulation Study on the Layered Phenomenon of Lakes and Reservoirs Destroyed by a Forced Circulation Device"

_jmse, doi:10.3390/jmse10050665_

Round 1
Reviewer 1 Report
Dear authors,
Congratulations for your article which is clearly presented and scientific soundness is well mature. At this stage, I think it can be published.
Good luck
Author Response
We are thankful for the reviewer’s comment.

Reviewer 2 Report
Originality and motivation can be clearly understood. However, the description and presentation especially regarding to modeling and figures are not sufficient (a part of those are described below), I could not judge whether the method and conclusion are appropriate. I would like to recommend to authors that the article will be improved and submitted again.
- turbulent equation based on k-e model
Author introduce Eq. (4) and (5) as a standard equation of k-e model. However, it is the first time that I see the expression. I could not understood the physical interpretations of Eqs. For an instance, in usual expression of k-e model, production term of k should be included in the right hand side of Eq.(4). Which term in Eq.(4) works for energy production? Furthermore, I guess that the equation must take account of turbulent energy consumption around the thermocline (interface between two different layers). Is that influence of stratification expressed by Eq. (4) and (5)?
- transport equation of temperature
Although the numerical results of water temperature are shown in sec.3, there are no any description of the equation to be solved for water temperature.
- modeling of density-change
In analysis regarding to density stratification, model of change of water density should be a crucial point. However, I could not find any information that explains how to evaluate and how to take account of the influence of difference of water temperature.
- representativeness of airlift device
In the section 2.3, the specification of airlift is induced. Is the assumed-specification of airlift representing a typical feature of devices? It should be needed to explain why the specification is assumed.
- Unclear and incomplete of figures
Many figures seems to be unclear and incomplete;
Figure 1 (b): figure is not clear and I did not distinguish structure of airlift device (which is guide, pipe, inlet of air?).
Figure 3 : Unit is not shown in legend ([m/s]?). Because there are no label appended to each panel, readers can not see the spatial-relationship between a stratification and airlift device. Furthermore, the location of airlift should be shown in panel.
Figure 4 : same as Figure 3. There are no label and unit.
Figure 5,6,7 : Similar markers are used in water-temperature and I could not distinguish.
Figure 8 : same as Figure 3. There are no label and unit.
Author Response
Response: We are thankful for the reviewer’s comment and suggestion.
Specific comments:
Point 1: turbulent equation based on k-e model
Author introduce Eq. (4) and (5) as a standard equation of k-e model. However, it is the first time that I see the expression. I could not understood the physical interpretations of Eqs. For an instance, in usual expression of k-e model, production term of k should be included in the right hand side of Eq.(4). Which term in Eq.(4) works for energy production? Furthermore, I guess that the equation must take account of turbulent energy consumption around the thermocline (interface between two different layers). Is that influence of stratification expressed by Eq. (4) and (5)?
Response 1: We are thankful for the reviewer’s comment. The purpose of this study is not to use numerical simulation methods, but to use Fluent software to evaluate the possibility of air-lift device on lake water cycle. Therefore, we accidentally added unnecessary numerical simulation theory to this paper, which brought reading trouble to the reviewers. So sorry. And that part has been removed. (Line 109~150).
Point 2: transport equation of temperature
Although the numerical results of water temperature are shown in sec.3, there are no any description of the equation to be solved for water temperature.
Response 2: Please refer to Response 1.
Point 3: modeling of density-change
In analysis regarding to density stratification, model of change of water density should be a crucial point. However, I could not find any information that explains how to evaluate and how to take account of the influence of difference of water temperature.
Response 3: Please refer to Response 1.
Point 4: representativeness of airlift device
In the section 2.3, the specification of airlift is induced. Is the assumed-specification of airlift representing a typical feature of devices? It should be needed to explain why the specification is assumed.
Response 4: The material and size of air lift affect the circulation efficiency of lake water, and it is generally believed that the larger the device, the better the effect. However, when the smaller size can achieve the desired effect, it will also be the most economical choice. Therefore, the specifications selected in this study draw on the research experience of peer experts. Through the simulation results, we can also verify that the specifications we set are appropriate.
Point 5: Unclear and incomplete of figures
Many figures seems to be unclear and incomplete;
Figure 1 (b): figure is not clear and I did not distinguish structure of airlift device (which is guide, pipe, inlet of air?).
Response 5.1: The geometry and configuration for the airlift device is shown in modified Fig. 1. (Line 175).
Figure 3 : Unit is not shown in legend ([m/s]?). Because there are no label appended to each panel, readers can not see the spatial-relationship between a stratification and airlift device. Furthermore, the location of airlift should be shown in panel.
Response 5.2: The label and unit were added to the Fig.3.(Line 236)
Figure 4 : same as Figure 3. There are no label and unit.
Response 5.3: The label and unit were added to the Fig.4. (Line 292)
Figure 5,6,7: Similar markers are used in water-temperature and I could not distinguish.
Response 5.4: The Figure was replaced. (Line 317~325)
Figure 8 : same as Figure 3. There are no label and unit.
Response 5.5: The label and unit were added to the Fig.8. (Line 363)
Reviewer 3 Report
The paper entitled “Numerical simulation study on layered phenomenon of lake 2 and reservoir destroyed by forced circulation device Research 3 on the destratification of lakes and reservoirs” uses computational fluid dynamics 16 method combined with Eulerian two-phase flow model to predict the damage of airlift to the thermal stratification phenomenon of lake water.
The paper propose novel ideas but some issues should be introduced in other to go further.
Comments:
1º) Title is heave. Authors should look for a fancier title.
2º) Equations from 1 to 8 seems to be pasted from other editor. Please improve visual aspect of the equations.
3º) l-144 the word “and” should be in text mode.
4º) Paragraph 159 suggests that you solve the Equations with FEMs or some equivalent numerical method. Give more details about the numerical method employed. Do you use a cubic-mesh? The author should organize it in a table in order to ease a future comparison. Do you use any initialization?
5º) l-182-> 10-3 should be a superscript.
6º) Figure 6 has labels unligned. Please format accordingly centered.
Author Response
Response: We are thankful for the reviewer’s comments and suggestions.
Comments:
Point 1: Title is heave. Authors should look for a fancier title.
Response 1: The title was changed to “Numerical simulation study on layered phenomenon of lake and reservoir destroyed by forced circulation device”.
Point 2: Equations from 1 to 8 seems to be pasted from other editor. Please improve visual aspect of the equations.
Response 2: We are thankful for the reviewer’s comments. The purpose of this study is not to use numerical simulation methods, but to use Fluent software to evaluate the possibility of air-lift device on lake water cycle. Therefore, we accidentally added unnecessary numerical simulation theory to this paper, which brought reading trouble to the reviewers. So sorry. And that part has been removed. (Line 109~150).
Point 3: l-144 the word “and” should be in text mode.
Response 3: Please refer to Response 2.
Point 4: Paragraph 159 suggests that you solve the Equations with FEMs or some equivalent numerical method. Give more details about the numerical method employed. Do you use a cubic-mesh? The author should organize it in a table in order to ease a future comparison. Do you use any initialization?
Response 4: The purpose of this study is not to use numerical simulation methods, therefore, we accidentally added unnecessary numerical simulation theory to this paper, which brought reading trouble to the reviewers. So sorry. And that part has been removed. (Line 109~150). The geometry and configuration for the airlift device is shown in modified Fig. 1. (Line 179).
Point 5: l-182-> 10-3 should be a superscript.
Response 5: The change of air viscosity is not considered in this paper, so this part is deleted. (Line 187).
Point 6: Figure 6 has labels unligned. Please format accordingly centered.
Response 6: The labels were modified.(Line 326).

Round 2
Reviewer 2 Report
This paper aims to unveil an influence of design and specification of airlift device to breaking a thermal-stratification in a lake. By conducting a series of computer fluid simulation with different conditions, it was found that the both device-size and inlet-gas-velocity affect liquid-velocity generated by the device, and a guide plate installed to the device can have a greater impact. The author well responded to the reviewer's comments and the paper seems to be improved appropriately. Because the background, methodology and results are clearly shown in the manuscript, and the findings of the paper probably can provide an effective basis for future field scale-up experiments, I recommend that the paper should be published in the journal.
Reviewer 3 Report
The paper has been improved. The techniques/method descriptions is now more clear.